



# Development of bacterial communities in biological soil crusts along a revegetation chronosequence in the Tengger Desert, northwest China

*Author names and affiliations:*

Lichao Liu[1], Yubing Liu[1,2] *, Peng Zhang[1], Guang Song[1], Rong Hui[1], Jin Wang[1,2]

[1]Shapotou Desert Research & Experiment Station, Northwest Institute of Eco-Environment and Resources, Chinese

Academy of Sciences, Lanzhou, 730000, China

[2]Key Laboratory of Stress Physiology and Ecology in Cold and Arid Regions of Gansu Province, Northwest Institute

of Eco–Environment and Resources, Chinese Academy of Sciences, Lanzhou 730000, China

* *Corresponding author:* Yubing Liu

Address: Donggang West Road 320, Lanzhou 730000, P. R. China.

Tel: +86 0931 4967202.

E-mail address: liuyb@lzb.ac.cn

**Abstract.** Knowledge of structure and function of microbial communities in different successional stages of biological soil crusts (BSCs) is still scarce for desert areas. In this study, Illumina MiSeq sequencing was used to assess the composition changes of bacterial communities in different ages of BSCs in the revegetation of Shapotou in the Tengger Desert. The most dominant phyla of bacterial communities shifted with the changed types of BSCs in the successional stages, from Firmicutes in mobile sand and physical crusts to Actinobacteria and Proteobacteria in BSCs, and the most dominant genera shifted from *Bacillus*, *Enterococcus* and *Lactococcus* to RB41_norank and JG34-KF-361_norank. Alpha diversity and quantitative real-time PCR analysis indicated that bacteria richness and abundance reached their highest levels after 15 years of BSC development. Redundancy analysis showed that soil pH, silt content and carbon:nitrogen ratio were closely related to the bacterial communities of BSCs. The results suggested that bacterial communities of BSCs recovered quickly with the improved soil physicochemical properties in the





early stages of BSC succession. Change in the bacterial community structures may be an important
indicator in the biogeochemical cycling and nutrient storage in early successional stages of BSCs in
desert ecosystems.
**Key words** biological soil crusts (BSCs), successional stages, bacterial community, revegetation,
desert ecosystem

## 1 Introduction

Biological soil crusts (BSCs) are assemblages of cryptogamic species and microorganisms, such

as cyanobacteria, green algae, diatoms, lichens, mosses, soil microbes and other related
microorganisms that cement the surface soil particles through their hyphae, rhizines/rhizoids and
secretions (Eldridge and Greene, 1994; Li, 2012; Pointing and Belnap, 2012; Weber et al., 2016).
Due to their specialized structures and complicated assemblages of their members, BSCs constitute
one of the most important landscapes and make up 40 % of the living cover of desert ecosystems,
even exceeding 75 % in some special habitats (Belnap and Eldridge, 2003). It is well known that
BSCs play critical roles in the structure and function of semi-arid and arid ecosystems (Eldridge and
Greene, 1994; Li, 2012). They contribute to ecological services such as soil stabilization, reduction
of wind and water erosion, and facilitation of higher plant colonization (Belnap, 2003; Belnap and
Lange, 2001; Maier et al., 2014; Pointing and Belnap, 2012). BSCs generally experience the main
successional stages in desert ecosystems: mobile sand, algal crust, lichen crust and moss crust (Lan
et al., 2012a; Liu et al., 2006). The different successional stages of BSCs vary in their ecological
function (Belnap, 2006; Bowker and Belnap, 2007; Li, 2012; Moquin et al., 2012).

Bacteria are the most abundant microorganisms and play important roles in the development

process of BSCs (Bates et al., 2010; Green et al., 2008; Gundlapally and Garcia-Pichel, 2006). They
can decompose organic material and release nutrients, mediating geochemical processes necessary
for ecosystem functioning in the persistence of BSCs (Balser and Firestone, 2005). Species
composition and community structure of bacteria change greatly during the successional process of
BSCs (Gundlapally et al., 2006; Moquin et al., 2012; Zhang et al., 2016). Most research on
prokaryotic diversity of BSCs has focused on cyanobacteria-dominated biocrusts in arid and semi-
arid regions (Abed et al., 2010; Garcia-Pichel et al., 2001; Nagy et al., 2005; Steven et al., 2013;
Yeager et al., 2004). Recent studies of the bacterial community structure of bryophyte- or lichen-



dominated crusts indicate that lichen-associated communities encompass a wide taxonomic
diversity of bacteria (Bates et al., 2011; Cardinale et al., 2008; Maier et al., 2014). Heterotrophic
bacteria may perform a variety of roles such as nutrient mobilization and nitrogen (N) fixation and
could be of considerable importance for the stability of lichen-dominated soil communities.
However, there have been few studies on changes of bacterial diversity and their function in BSCs
during the development process in desert zones, and these only in the Sonoran (Nagy et al., 2005)
and Gurbantunggut Deserts (Zhang et al., 2016). What changes occur in bacterial community
composition and their roles in improving soil properties in different successional stages of BSCs?
What is the significance of these changes on BSC succession in the recovery process of desert
revegetation in temperate zones?

A recent study on crusts in the Tengger Desert, China, showed that bacterial diversity and

richness were highest after 15 years, and at least 15 years might be needed for recovery of bacterial
abundance of BSCs (Liu et al., 2017). To better understand these questions, we must analyze in
detail the bacterial community composition of BSCs at all levels of classification and their
corresponding function in the recovery process of BSCs. In the present study, bacterial community
composition and potential function were analyzed in BSCs along a chronosequence of over 50-year-
old revegetation. We hypothesized that bacteria are the key species in carbon (C) accumulation and
soil improvement in early stages of BSC succession.
**2 Materials and methods**
**2.1 Study site description**

The study site is located at Shapotou, southeast fringe of the Tengger Desert, northwest China.

The nature landscape is characterized by the reticulated chains of barchan dunes with the vegetation
cover less than 1%. The mean annual precipitation is about 180 mm with large seasonal and inter-
annual variation. The mean wind speed is 3.5 m/s, and the average days with dust events are 122 d
per year. The revegetation protection system for Bao–Lan railway in this area was established
initially in 1956, and was expanded in 1964, 1973, 1981 and later through the plantation of the
xerophilous shrubs. This unirrigated revegetation system works quite well to protect the railroad
line from sand bury and dust hazard during past sixty years. Also, the experimental plots of less
than one hectare were established with the same plantation techniques by the Shaptou desert





research and experiment station in 1987, 2000, and 2010 in the nearby sand dunes. These sand fixed
areas provide an ideal temporal succession sequence for studying the variation of environmental
factors following plantation in the floating sand. As mentioned in other literatures, the initial state
of BSCs began to form following the stabilization of sand dunes and developed with the colonization
of cryptogam (Liu, et al, 2006). The appeared BSCs can be divided into four types, such as physical
crusts, algal-dominated, lichen-dominated and moss-dominated crusts.   In this study, we selected
BSCs from the revegetation established in 1964, 1981, 1987, 2000 and 2010, and non-fixed mobile
sand as the control (Figure 1). BSCs were sampled in November 2015, and named according to the
fixed-sand time as 51YR (51 years of revegetation), 34YR, 28YR, 15YR, 5YR and MS, respectively.
The main types of BSCs were cyanobacteria–lichen- and moss-dominated crusts from 15YR to
51YR.

## 2.2 BSC sampling

In each revegetation, BSC samples were collected in early November 2015. Five soil cores (3.5-
cm diameter) with crust layers from four vertices of a square (20-m length) and a diagonal crossing
point in each plot (Figure 1 C)  were sampled individually using a sterile trowel. To decrease spatial
heterogeneity, each BSC sample was taken from six individual plots (at least 20 m between two
adjacent plots) from each revegetation time. Therefore, we obtained 30 BSC samples in total (5
cores × 6 individual plots) and these were mixed together to form one composite BSC sample.
Triplicate composite samples for each revegetation time were collected and the BSC samples were
preserved in an ice box. Samples were then taken back to the laboratory, immediately sieved (by 1
mm) to remove stones and plant roots, homogenized thoroughly and stored at –70 °C for subsequent
analyses.

## 2.3 DNA extraction and Illumina MiSeq sequencing

Microbial DNA was extracted from BSC samples using E.Z.N.A Soil DNA (Omega Bio-tek,
Norcross, GA, U.S.) according to the manufacturer's protocols. The extracted DNA was diluted in
TE buffer (10 mM Tris–HCl and 1 mM EDTA at pH 8.0) and stored at –20 °C until use. An aliquot
of the extracted DNA from each sample was used as a template for amplification. The bacteria 16S
ribosomal RNA gene was amplified by PCR (95 °C for 3 min, followed by 25 cycles at 95 °C for
30 s, 55 °C for 30 s and 72 °C for 45 s, and a final extension at 72 °C for 10 min) using primers





338F (5′-ACTCCTACGGGAGGCAGCA-3′) and 806R (5′-GGACTACHVGGGTWTCTAAT-3′).
PCRs were performed in triplicate 20-μL mixture containing 2 μL of 5 × FastPfu Buffer, 2 μL of
2.5 mM dNTPs, 0.8 μL of each primer (5 μM), 0.2 μL of FastPfu Polymerase and 10 ng of template
DNA. This was conducted according to Wang et al. (2015). Amplicons were extracted from 2 %
agarose gels and purified using the AxyPrep DNA Gel Extraction Kit (Axygen Biosciences, Union
City, CA, USA) according to the manufacturer's instructions and quantified using QuantiFluor™ -
ST (Promega Corporation, Madison, WI, USA).

Purified amplicons were pooled in equimolar and paired-end sequenced (2 × 300) on an Illumina

MiSeq platform according to the standard protocols at Majorbio Bio-Pharm Technology Co. Ltd.,
Shanghai, China (http://www.majorbio.com). The raw reads were deposited in the NCBI Sequence
Read Archive database (Accession number: SRP091312).

## 2.4 Quantitative real-time PCR (qPCR)

qPCR was performed to determine the absolute 16S rRNA gene abundance. We used the primer

sets of 515F (5′-GTGCCAGCMGCCGCGGTAA-3′) and 806R to quantify the total bacterial
populations. The standard templates were made from 10-fold dilutions of linearized plasmids
containing the gene fragment of interest that was cloned from amplified pure culture DNA. The 20
μL reaction mixtures contained 10 μL of 2 × SYBR Mix (with ROX) (DBI Bioscience,
Ludwigshafen, Germany), 0.4 μL each of 10 μM forward and reverse primers, 1 μL of total DNA
template (1 ng/μL) and 8.2 μL of RNase-free ddH$_2$O. The reaction was conducted on a Stratagene
Mx3000P Real-time PCR system (Stratagene, Agilent Technologies Inc., Santa Clara, CA, USA)
using the following program: 94 °C for 3 min followed by 40 cycles of 94 °C for 30 s, 58 °C for 30
s and 72 °C for 30 s, then 72 °C for 2 min. The detection signal was collected at 72 °C for 30 s and
analyzed. The melting curve was obtained to confirm that the amplified products were of the
appropriate size. For each soil sample, the qPCRs were repeated six times.

## 2.5 Processing of sequencing data

Raw fastq files were demultiplexed, quality-filtered using QIIME (version 1.17) with the

following criteria: (i) The 300-bp reads were truncated at any site receiving an average quality score
< 20 over a 50-bp sliding window, discarding the truncated reads shorter than 50 bp; (ii) exact
barcode matching, two nucleotide mismatch in primer matching, reads containing ambiguous





characters were removed and (iii) only sequences that overlapped > 10 bp were assembled according
to their overlap sequence. Reads that could not be assembled were discarded.
Operational taxonomic units (OTUs) were clustered with 97 % similarity cut-off using UPARSE
(version 7.1 http://drive5.com/uparse/) and chimeric sequences were identified and removed using
UCHIME. The taxonomy of each 16S rRNA gene sequence was analyzed by RDP Classifier
(http://rdp.cme.msu.edu/) against the SILVA (SSU115) 16S rRNA database using a confidence
threshold of 70 %. Hierarchical clustering analysis was performed using CLUSTER and visualized
using TREEVIEW, and other statistical analyses were performed with the IEG pipeline
(http://ieg.ou.edu). The average data were calculated for BSCs of each revegetation before analyzing
the unique and shared OTUs/genera. The figures were generated with OriginPro 9.1 and Excel 2013.
Alpha-diversity analysis was used to reflect the richness and diversity of microbial communities. In
order to investigate the overall differences in community composition among the samples, principal
component analysis (PCA) was performed using unweighted UniFrac distance (Lozupone and
Knight, 2005). Redundancy analysis (RDA) was used to assess the relationship between bacterial
compositions of BSCs and top soil physicochemical properties by permutation test analysis (Zhang
et al., 2016). Phylogenetic analysis of the top abundance genus were aligned with closely related
16S rRNA gene sequences, previously selected according to initial BLAST analyses and
downloaded from the NCBI website (http://www.ncbi.nlm.nih.gov), using CLUSTAL W
(Gundlapally and Garcia-Pichel, 2006). Phylogenetic trees were constructed using approximately-
maximum-likelihood routine by FastTree (version 2.1.3 http://www.microbesonline.org/fasttree/).

## 3 Results

### 3.1 Overview of sequencing and bacterial diversity

Illumina MiSeq sequencing was used to assess the bacterial community composition and
diversity of BSCs in successional stages for revegetation in Shapotou. Total 18 libraries of bacterial
16S rRNA were constructed, at least 37,332 effective sequences in each sample were obtained, and
an average length of 437 bp. 1197–2307 OTUs were generated using a threshold of 0.97 (Table S1).
394 OTUs were shared and occupied a relatively high proportion among all samples (17.07–32.92 %)
(Table S2), and these OTUs accounted for 41.96–84.88 % of the total sequences (Table S2). This
indicated a high coherence of community among these soil crusts. Alpha-diversity analysis revealed




the microbial richness and diversity. Rarefaction curves showed that the most bacterial OTUs were
found in 51YR crust, whereas MS contained the fewest. The number of OTUs was almost the same
from 15YR to 51YR (Figure 2). Community richness estimation using ACE and Chao revealed a
similar trend to that for community diversity, which was further supported by Shannon's indexes
(Table S1). Hierarchical clustering analysis (Figure 3 A) and PCA (Figure 3 B) showed that the
triplicate samples of each age of BSCs were clustered, verifying that the sequencing results were
reliable and the samples were reproducible.

**3.2 Bacterial community composition at high taxonomic levels**

In the bacterial community, a total of 28 phyla were retrieved at genetic distances of 3 %, and
they clustered into four groups according to their relative abundance (Figure 4). Of the total
sequences, 4.48 % were not classified at the phylum level. The percentages of major phyla for each
age of BSCs are shown in Figure 5. The most abundant phylum shifted from Firmicutes (72.8 %)
in MS and 5YR to Actinobacteria in BSCs (minimum 27.4 % in 15YR and maximum 30.7 % in
51YR). The following major phyla were at high abundance (> 10 % of total OTUs): Proteobacteria,
Chloroflexi, Acidobacteria and Cyanobacteria. The low-abundance phyla (1 % < of total OTUs <
10 %) were Gemmatimonadetes, Bacteroidetes, Armatimonadetes, Verrucomicrobia and
Deinococcus-Thermus. The percentages of Proteobacteria, Chloroflexi and Acidobacteria were
nearly the same after 15 years of development of BSCs. Cyanobacteria, in addition to the high
proportion for 15YR (16.13 %), also had a high proportion in 51YR (9.32 %). The other 17 phyla
were all < 1 % of total OTUs and so were removed from further analysis.
At the class level (Table 1), 95.61 % of sequences were assigned, and there was considerable
consistency in dominant classes among the crusts. Bacilli was the largest class in MS and 5YR with
sequence percentages of 68.73 and 32.62 %, respectively; and Actinobacteria was the predominant
class from 15YR to 51YR. In addition to subdivisions of Proteobacteria, other major classes
included Acidobacteria, Cyanobacteria, Chloroflexi, Clostridia, Cytophagia, Deinococci,
Gemmatimonadetes, Ktedonobacteria, Sphingobacteria and Thermomicrobia. The percentages of
high (> 10 % of total OTUs) and low abundance (1 % < of total OTUs < 10 %) classes decreased
from 98 % in MS to 89.29 % in 51YR, and minor and unclassified classes increased from 1.96 %
in MS to 10.67 % in 51YR.



At the family level, there were 133 identified families (data not shown), with the most abundant
families being Bacillaceae, Enterococcaceae and Streptococcaceae (Table S3). Other dominant
families were Geodermatophilaceae, JG34-KF-161, JG34-KF-361, Methylobacteriaceae,
Micromonosporaceae, Bradyrhizobiaceae and Enterobacteriaceae.
**3.3 Characterization of major genera and species**
A large proportion of sequences were not assigned to any genera. Even for genera with relative
abundance > 1 % in any samples, unclassified sequences occupied a high proportion (4.87–8.59 %).
Moreover, higher percentages of total sequences (from 13.51 % in MS to 37.28 % in 51YR) were
found in low-abundance genera (< 1 % in any samples) (Table S4). A total of 460 genera were found
in the crusts, of which 201 were shared by all BSC samples (data not shown). The major genera in
each age of BSCs are summarized in Figure 6. *Bacillus*, *Enterococcus* and *Lactococcus* were the
primary genera and represented 64.31 % of the total sequences in MS, and decreased to 30.20 % in
5YR and only 2.63 % in 51YR, indicating that these three genera were predominant in mobile sand
or physical crusts. Enterobacteriaceae_unclassified and *Alkaliphilus* were low-abundance genera in
MS. With the decrease in the three primary genera from MS to 51YR, a series of genera increased
in BSCs compared with MS and 5YR, including RB41_norank, JG34-KF-361_norank,
Acidimicrobiales_uncultured, JG34-KF-161_norank, JG30-KF-CM45_norank, *Microvirga*,
Actinobacteria_norank and *Rubrobacter* (relative abundance > 2 %).
The phylogenetic relationships of the 30 most abundant genera are shown in Figure 7. They
clustered into three groups at the phylum level: Actinobacteria formed one group and included 10
genera; another group was Firmicutes and Proteobacteria; and Cyanobacteria, Chloroflexi and
Deinococcus-Thermus formed the third group. The genera *Bryobacter* and *Blastocatella* in phylum
Acidobacteria were divided into two different groups.
*Bacillus* was the primary genus and represented 31 % sequences in MS (Table S4). An
unclassified species in this genus reached nearly 30 % relative abundance in MS (Figure 8). In the
*Enterococcus* genus, another core component, there was also an unclassified species with high
abundance. In the core species (Figure 8), *Bacillus*_unclassified, *Enterococcus*_unclassified,
*Lactococcus_piscium*, Enterobacteriaceae_unclassified and *Alkaliphilus_oremlandii*_OhILAs were
predominant and decreased from MS to 51YR; only *Acidimicrobiales*_unclassified increased, and



this represented the highest proportion in 51YR (2.62 %). The relative abundance of the primitive
species in MS and physical crusts decreased in BSCs (from 15YR to 51YR) because of the increased
numbers of species. There was little difference in numbers of genera and species among biocrusts
(from 15YR to 51YR), only in sequence numbers.

**3.4 Relationships between bacterial community structure and soil**

**physicochemical properties**

RDA (Figure 9) and hierarchical clustering analysis (Figure 3) were used to discern the
correlations between bacterial communities and soil physicochemical properties. The BSC the
grouping patterns of bacterial communities at the phylum and genus levels were similar to the OTU
level, with all divided into two groups. Group I contained two members, MS and 5YR, which
dominated the physical crusts and cyanobacterial crusts (Figure 1 A and B), and had the lowest
diversities with Shannon indexes of 3.3 and 4.61, and Simpson indexes of 0.139 and 0.0531,
respectively (Table S1). The remaining BSCs comprised the largest branch of Group II, which
dominated BSCs composed of algae, lichens or mosses (Figure 1 C–F), and had higher diversity
with Shannon indexes > 6.0 (Table S1).
From Figure 9, it can be inferred that BSC development was associated with soil
physicochemical properties (data from Li et al., 2007a; Table S5). The development of microbial
community structure was positively correlated with the physicochemical index except for soil bulk
density. Thirteen soil physicochemical variables were all significant testified by the permutation test
analysis ($p < 0.05$): total water content; pH; C:N ratio; silt and clay content; organic C; $CaCO_3$; total
phosphorus (P), nitrogen (N), potassium (K) and salt; electrical conductivity (EC) and maximum
water-holding capacity (WHC). Among them, soil pH, C:N ratio and silt content were the most
influential variables (Fig. 9).

**3.5 Quantification of bacterial abundance**

The averaged bacterial abundance in MS was $1.12 \times 10^6$ copies (16S rRNA gene) per gram of
soil (Table 2). Similar to the shift of bacterial richness, gene copies increased quickly in the initial
15 years of BSC development, and reached the approximate highest level of $2.70 \times 10^8$ copies in
15YR. There were no significant differences among 28YR, 34YR and 51YR.

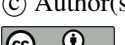



**4 Discussion**

Due to the species concept is relatively well-defined in BSC organisms, BSCs may act as a useful model system for diversity-function research. Their functional attributes are relatively well-known and estimation and manipulation of biodiversity in experiments are feasible, at least within some groups of BSC biota (Bowker et al., 2010). This relationship is more easily interpreted in artificially-constructed BSCs. During successional stages of BSCs, physical crusts in mobile sand contain the lowest C and N contents (Zhang et al., 2009). Algal crust is the earliest biocrust stage. It shows a surface thin layer which composed by aeolian-born materials and an organic layer formed by filamentous cyanobacteria associated with sand particles (Housman et al., 2006; Zhang, 2005; Zhang et al., 2009). Lichen and moss appear following with stabilization of the algal filaments on the soil surface. The C and N fixation rates are increased in lichen crust (Evans and Lange, 2003; Lan et al., 2012b; Zhang et al., 2010), and there is higher photosynthesis, exopolysaccharide and nitrogenase activity in moss crust compared to the early successional crusts (Housman et al., 2006; Lan et al., 2012b). In the successional process of BSCs, the microbial composition and community structure change greatly (Hu and Liu, 2003; Zhang et al., 2009). Crust succession is positively correlated with phospholipid fatty acid content and microbial biomass (Liu et al., 2013). The microbial biomass of soils is the most important driving force in most terrestrial ecosystems, largely due to control of conversion rates and mineralization of organic matter (Albiach et al., 2000; Baldrian et al., 2010). Bacteria have a highest proportion of the microbial biomass in soils (Maier et al., 2014; Wang et al., 2015), and thus have important roles in the successional process of BSCs.

**4.1 Impact of BSC age on bacterial community composition**

In the present study, we gained information concerning the diversity of bacterial communities in BSCs of different ages in restored vegetation at Shapotou in the Tengger Desert. The 16S rRNA gene-based amplicon survey revealed the dominance of Actinobacteria, Proteobacteria, Chloroflexi, Acidobacteria and Cyanobacteria in all BSCs, with Firmicutes dominating MS (72.8 %) and decreasing to 3.05 % in 51YR, and Actinobacteria increasing from 15YR (27.4 %) to 51YR (30.7 %). Due to different arid conditions, comparisons with other studies of BSCs should be viewed with caution. Cyanobacteria, Actinobacteria, Proteobacteria and Acidobacteria are ubiquitous in soils and sediments everywhere, in arid as well as wet landscapes (Fierer et al. 2012), and Proteobacteria



are very common and diverse among all BSCs. We observed that Actinobacteria were the most
abundant phylum in the developing (15YR, 28YR and 34YR) and relatively developed (51YR)
BSCs, similar to BSCs from the Colorado Plateau and the Sonoran Desert, where Actinobacteria
were dominant (Gundlapally and Garcia-Pichel 2006; Nagy et al. 2005; Steven et al. 2013).
Actinobacteria and Proteobacteria are usually predicted to be copiotrophic groups which increase
in high C environments (Fierer et al., 2007). These results differ from those reported in BSCs from
Oman and the Gurbantunggut Desert (Abed et al. 2010; Moquin et al., 2012; Zhang et al., 2016),
and even from BSCs of natural vegetation at the edge of the Tengger Desert (Wang et al., 2015),
where Proteobacteria were the most abundant phylum followed by Cyanobacteria, Actinobacteria
and Chloroflexi. Unexpectedly, Cyanobacteria had a high proportion in the developed BSCs,
although they were prevalent in early successional stages of BSCs (5YR) and play crucial roles in
initial crust development (Belnap and Lange, 2001). This is relatively similar to that in the natural
habitat around the Tengger Desert, where Cyanobacteria (19.5 %) and Actinobacteria (19.4 %) were
the most dominant phyla after Proteobacteria (25.0 %). Moreover, the results did not resemble those
from arid Arizona soils (Dunbar et al., 1999) or the Gurbantunggut Desert (Zhang et al., 2016) due
to the high proportion of Chlorflexi, an unexplained presence of thermophilic phyla (Gundlapally
and Garcia-Pichel, 2006; Moquin et al., 2012; Nagy et al., 2005) displays good adaptation to drought
environment and important roles in the development of BSCs in arid zones (Lacap et al., 2011;
Wang et al., 2015).

## 4.2 Function of BSC bacteria

More and more information about BSC bacteria has been reported with the convenience of

culture-independent sequencing methods, and studies of their function and classification in BSCs
are increasingly detailed. The main function of these dominant bacteria involves the cycling and
storage of C and N in desert ecosystems, which is vital to functioning of arid land (Weber et al.,
2016). Firmicutes are more frequently detected in below-biocrust soils (1–2 cm depth) (Elliott et al.,
2014) and dominated in MS and 5YR, with the vast majority of abundant species being in Firmicutes
in the Tengger Desert. Cyanobacteria are the main contributors to C and N fixation in soils during
successional processes of BSCs (Belnap and Gardner, 1993). They are thought to serve as pioneers
in the stabilization process of soils (Garcia-Pichel and Wojciechowski 2009), of which genus



*Phormidium* is significantly more abundant in surface soils (0–1 cm depth), and genus *Microcoleus*
is globally dominant as biocrust-forming microorganisms in most arid lands and their production of
polysaccharide sheaths aids in formation of cm-long filament bundles (Belnap and Lange 2003;
Boyer et al. 2002; Garcia-Pichel et al. 2001; Pointing and Belnap 2012). In addition to the
filamentous bacteria of *Microcoleus* and *Phormidium*, *Mastigocladopsis* and *Trichocoleus* were
also in the 30 most abundant genera of BSCs in Shapotou, and mainly harvest energy from light.
*Pseudonocardia*, a mycelial genus of Actinobacteria, were dominant and are likely important during
BSC formation (Weber et al., 2016). Proteobacteria and Bacteroidetes can produce
exopolysaccharides, so they could also play roles in soil stabilization and BSC formation
(Gundlapally and Garcia-Pichel 2006). Owing to limited culture collections and curated sequence
databases of BSC bacteria, most non-cyanobacterial sequences from DNA-based bacterial surveys
cannot be reliably named or taxonomically defined, especially in relatively abundant genera in
Actinobacteria and Proteobacteria, such as *Bosea*, *Microvirga*, *Rubellimicrobium*, *Patulibacter*,
*Solirubrobacter*, *Blastococcus* and *Arthrobacter* in the present study. Discovery and
characterization of the functions of these dryland-adapted bacteria is a challenging area for future
study.
**4.3 Relationship between bacterial community shift and soil physicochemical**
**properties**
PCA and RDA showed that bacterial community compositions of MS and 5YR significantly
differed from those of BSCs of more than 15 years in age, and were positively correlated with soil
physicochemical properties. Combined with the results of alpha-diversity analysis and qPCR, this
means that the species richness and abundance reached their highest levels at 15 years of BSC
development and then maintained similar levels thereafter. Similar trends were found in recovery
of soil properties and processes after sand-binding at five different-aged revegetated sites –
proportions of silt and clay, depth of topsoil and concentrations of soil K, total N, total P and organic
C increased with years since revegetation (Li et al., 2007a, b). The annual recovery rates of soil
properties was greater in the initial revegetated sites (0–14 years) than that in the old revegetated
sites (43–50 years) (Li et al., 2007a). These results suggest that bacterial communities of BSCs
recovered quickly in the fastest recovery phase of soil properties (the initial 15 years), and the

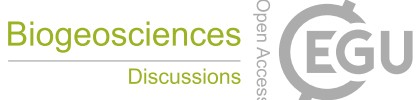

bacterial biomass increased with the improvement of soil texture and nutrients, especially pH, C:N
ratio, silt content and total P and K in the Tengger Desert. This may be attributed to vegetation
composition, soil temperature and soil moisture, because they are key factors regulating soil
microbial composition and activity (Butenschoen et al., 2011; De Deyn et al., 2009; Sardans et al.,
2008), soil nutrient uptake and release (Peterjohn et al., 1994; Rustad et al., 2001), especially in the
BSCs of top soil. BSC, plant and soil biochemical properties together lead to microbial diversity of
BSCs in long-term revegetation, and the microorganisms in turn improve soil texture (Li et al.,
2007b, 2010).
**5 Conclusions**
Assessing of bacterial community structure by Illumina MiSeq sequencing showed that changes
of bacterial diversity and richness were consistent with the recovery phase of soil properties in
different successional stages of BSCs in the revegetation of Shapotou in the Tengger Desert. The
shift of bacterial community composition in BSCs at all levels of classification was related to their
corresponding function in the BSC recovery process. These results confirmed our hypothesis that
bacteria are key microorganisms in nutrition accumulation and soil improvement in early stages of
BSC succession.

*Data availability*. Raw data for Illumina MiSeq sequencing of 18 samples was deposited in the
NCBI Sequence Read Archive database (https://www.ncbi.nlm.nih.gov/sra/?term=SRP091312).

*Author contributions*. Yubing Liu and Lichao Liu designed the research. Peng Zhang, Guang Song
and Rong Hui collected samples from the field. Yubing Liu and Jin Wang performed DNA
extraction and quality detection. Yubing Liu analyzed the high-throughput data and prepared the
manuscript with consistent contributions from Lichao Liu.

*Competing interests*. The authors declare that they have no conflict of interest.

*Acknowledgments*. This work was financially supported by the Creative Research Group Program
of National Natural Science Foundation of China (grant No. 41621001) and the National Natural



Science Foundation of China (grant No. 41371100).

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




**Table 1**. Percentages of the major classes in each age of BSCs. MS, 5YR, 15YR, 28YR, 34YR and 51YR represent
mobile sand, 5, 15, 28, 34 and 51-year-old BSCs, respectively.

| Dominant | MS | 5YR | 15YR | 28YR | 34YR | 51YR |
|---|---|---|---|---|---|---|
| **Bacilli** | 68.73281 | 32.6217 | 10.87003 | 18.88014 | 14.65767 | 2.809922 |
| **Actinobacteria** | 10.25572 | 17.22651 | 27.36705 | 28.34208 | 29.31533 | 30.65824 |
| **Alphaproteobacteria** | 4.058181 | 12.26026 | 19.93375 | 16.30594 | 18.98282 | 21.11772 |
| **Acidobacteria** | 1.404514 | 2.372406 | 11.75488 | 8.32619 | 7.703847 | 9.022644 |
| **Chloroflexia** | 0.886639 | 2.423301 | 4.006393 | 2.962606 | 3.367977 | 3.857281 |
| **Cyanobacteria** | 0.112504 | 16.13272 | 3.943891 | 2.275974 | 2.367049 | 9.32444 |
| **Clostridia** | 4.091218 | 1.661666 | 0.517876 | 1.017893 | 0.704489 | 0.15447 |
| **Cytophagia** | 0.265188 | 1.223258 | 0.93039 | 0.739312 | 1.022358 | 1.579521 |
| **Deinococci** | 0.048216 | 1.255402 | 0.342869 | 0.372335 | 0.249116 | 0.20715 |
| **Deltaproteobacteria** | 0.447337 | 0.740205 | 1.150934 | 0.993785 | 1.087539 | 1.255402 |
| **Gammaproteobacteria** | 5.715383 | 2.632237 | 1.011643 | 1.890246 | 1.417015 | 0.425908 |
| **Gemmatimonadetes** | 0.645559 | 2.400979 | 2.406336 | 2.646523 | 2.75992 | 2.40455 |
| **Ktedonobacteria** | 0.053573 | 0.113397 | 1.75542 | 1.121469 | 2.072395 | 1.657202 |
| **Sphingobacteriia** | 0.262509 | 0.666095 | 1.200043 | 0.897353 | 0.995571 | 0.889317 |
| **Thermomicrobia** | 0.449123 | 1.351834 | 3.24208 | 3.414408 | 3.008143 | 2.810815 |
| **Betaproteobacteria** | 0.572342 | 0.789314 | 0.939319 | 1.021465 | 1.073253 | 1.11254 |
| **Minor** | 0.018688 | 0.039555 | 0.080851 | 0.08194 | 0.081753 | 0.085887 |
| **Unclassified** | 0.000911 | 0.00142 | 0.005018 | 0.005822 | 0.009866 | 0.02084 |



**Table 2**. Absolute abundances of bacteria (copies of ribosomal genes per gram of soil) in BSCs quantified by qPCR
(means ± standard deviation, n = 6). MS, 5YR, 15YR, 28YR, 34YR and 51YR represent mobile sand, 5, 15, 28, 34
and 51-year-old BSCs, respectively.

| Dominant | MS | 5YR | 15YR | 28YR | 34YR | 51YR |
|---|---|---|---|---|---|---|
| **Bacteria abundance** | $1.12 \times 10^6 \pm$ $4.19 \times 10^5$ a | $3.94 \times 10^7 \pm$ $2.21 \times 10^6$ b | $2.70 \times 10^8 \pm$ $1.91 \times 10^7$ c | $5.44 \times 10^8 \pm$ $4.23 \times 10^7$ c | $7.61 \times 10^8 \pm$ $8.5 \times 10^7$ c | $9.03 \times 10^8 \pm$ $2.55 \times 10^7$ c |

Means with different letters are significantly different ($P < 0.05$).










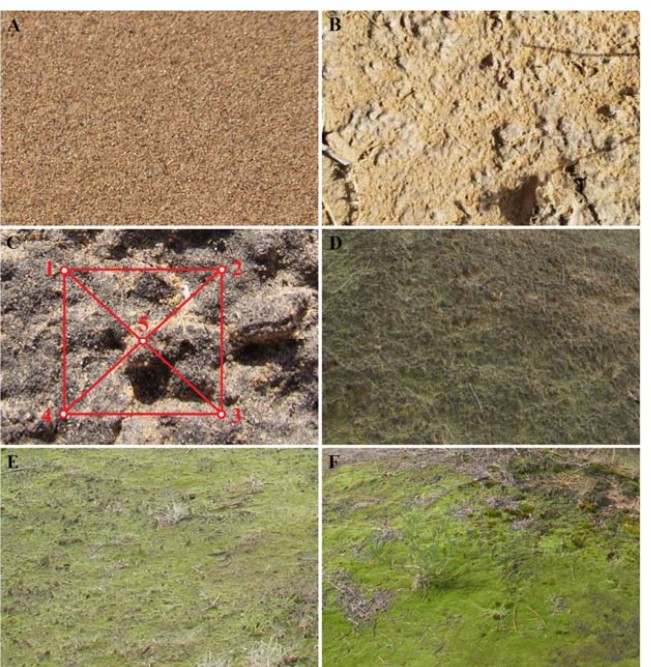


**Figure 1**. Sand dune landscape before (MS, A) and after establishing sand-binding vegetation with physical crusts
dominated by few cyanobacteria, revegetated in 2010 (5YR, B); with BSC dominated by cyanobacteria, revegetated
in 2000 (15YR, C); with BSC dominated by cyanobacteria and algae, revegetated in 1987 (28YR, D); with BSC
dominated by lichens, revegetated in 1981 (34YR, E); and with BSC dominated by mosses, revegetated in 1964
(51YR, F). Five soil cores (3.5-cm diameter) with crust layers from four vertices of a square (20-m length) and a
diagonal crossing point in each plot were sampled individually (as shown in C).

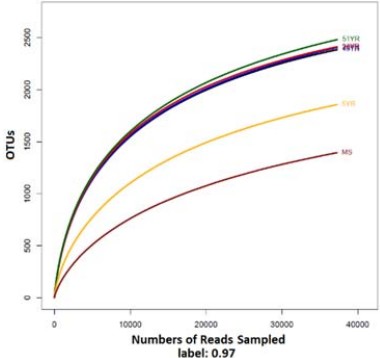


**Figure 2**. Rarefaction results of the 16S rDNA libraries based on 97 % similarity in different age of BSCs. MS, 5YR,
15YR, 28YR, 34YR and 51YR represent mobile sand, 5-, 15-, 28-, 34- and 51-year-old BSCs, respectively.



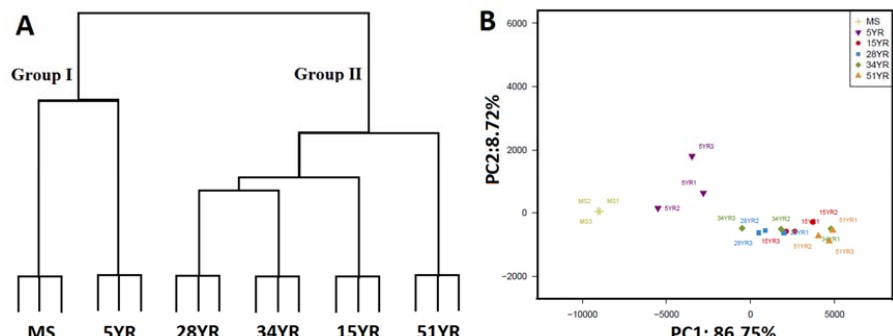


**Figure 3**. Hierarchical clustering analysis and PCA of bacterial communities in six different ages of BSCs at OTU


level based on 97 % similarity (triplicate samples for each age). MS, 5YR, 15YR, 28YR, 34YR and 51YR represent


mobile sand, 5-, 15-, 28-, 34- and 51-year-old BSCs, respectively.





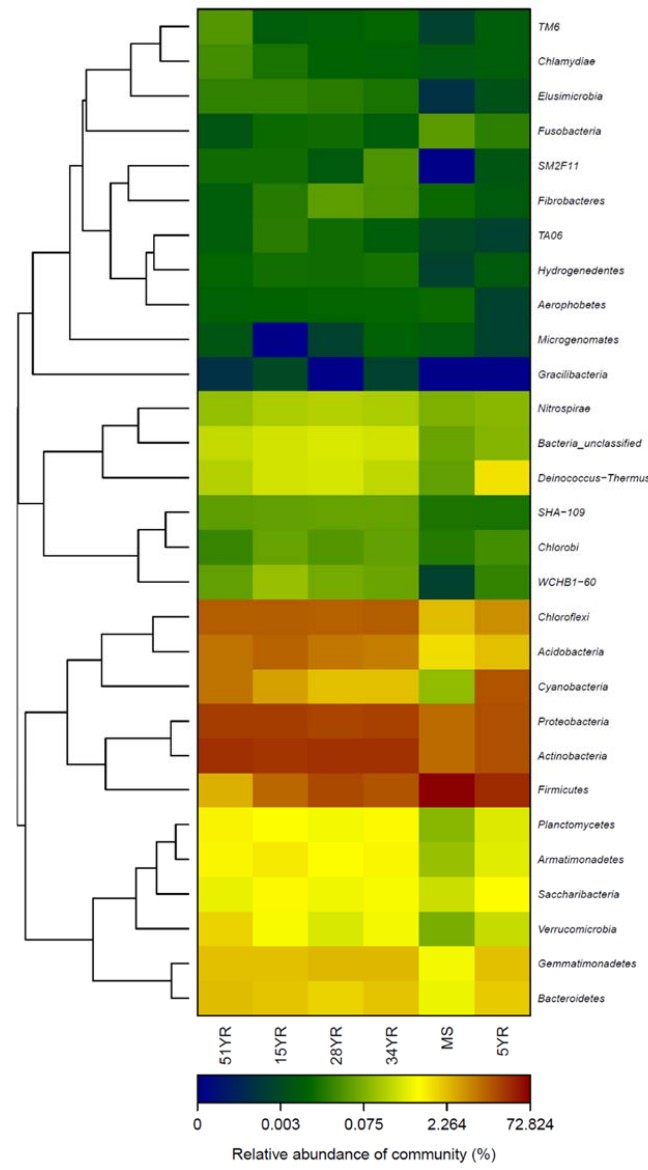


**Figure 4**. Heatmap of bacterial communities in different ages of BSCs at phylum level. MS, 5YR, 15YR, 28YR,

34YR and 51YR represent mobile sand, 5-, 15-, 28-, 34- and 51-year-old BSCs, respectively.



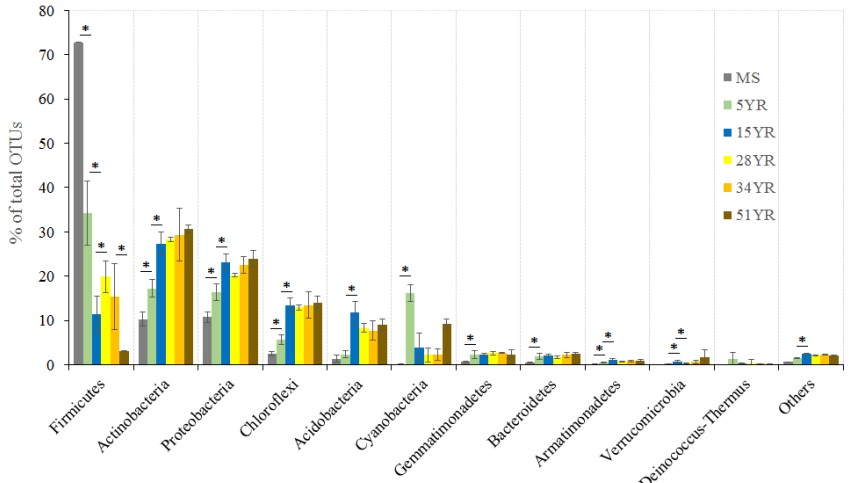


**Figure 5**. Abundant phyla (> 10 % of total OTUs) and low-abundance phyla (1 % < of total OTUs < 10 %) of

bacteria distributed in different ages of BSCs. Data are defined at a 3 % OTU genetic distance. Data are presented

as mean ± standard deviation; n = 3 per BSC sample. Paired t-test (BSC samples) was used to assess the significance

between adjacent ages of BSCs. *P ≤ 0.05, **P ≤ 0.001. MS, 5YR, 15YR, 28YR, 34YR and 51YR represent mobile

sand, 5, 15, 28, 34 and 51-year-old BSCs, respectively.


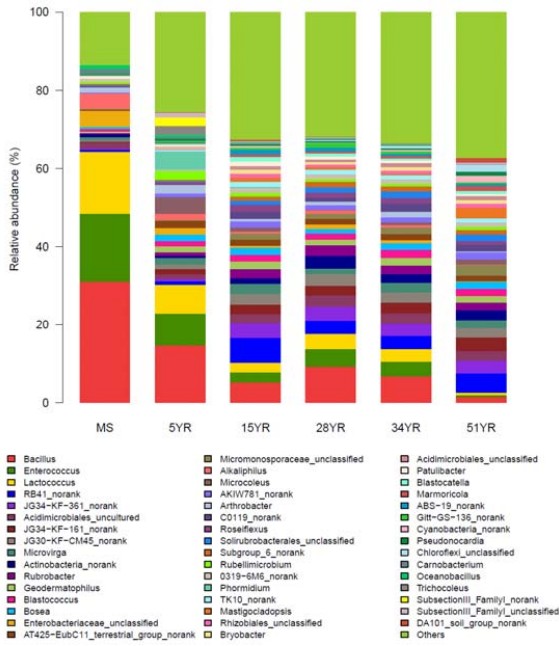


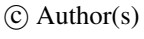



**Figure 6**. Bacterial community composition in six different ages of BSCs at the genus level. Data are defined at a
3 % OTU genetic distance. MS, 5YR, 15YR, 28YR, 34YR and 51YR represent mobile sand, 5, 15, 28, 34 and 51-
year-old BSCs, respectively.

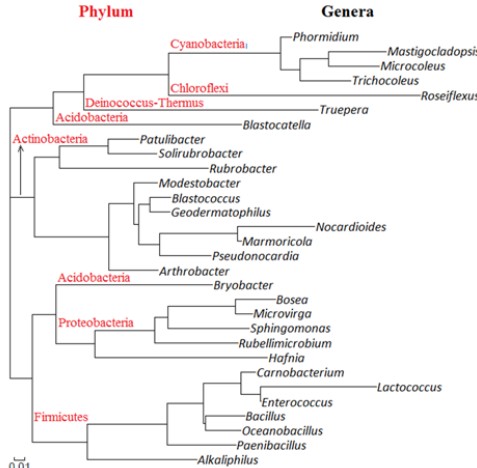


**Figure 7**. Phylogenetic relationship of the 30 most abundant genera in bacterial composition of BSCs.

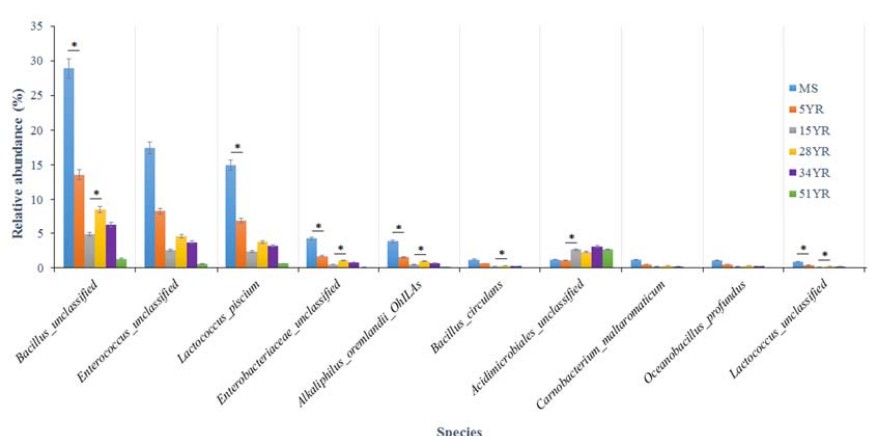


**Figure 8**. Abundant species (> 10 % of total OTUs) and low-abundance species (1 % < of total OTUs < 10 %) of
bacteria distributed in different ages of BSCs. Data are defined at a 3 % OTU genetic distance. Data are presented
as mean ± standard deviation; n = 3 per BSC samples; Paired t-tests (BSC samples) were used to assess the
significance between the adjacent ages of BSCs. *P ≤ 0.05, **P ≤ 0.001. MS, 5YR, 15YR, 28YR, 34YR and 51YR
represent mobile sand, 5, 15, 28, 34 and 51-year-old BSCs, respectively.

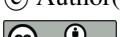



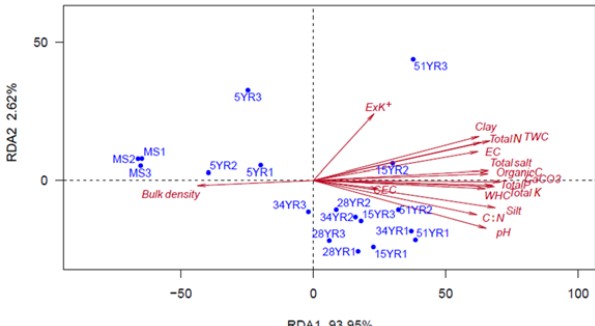


**Figure 9**. Redundancy analysis (RDA) of bacterial community structures in relation to soil physiochemical
properties. Arrows indicate the direction and magnitude of soil physiochemical index associated with bacterial
community structures. The length of arrows in the RDA plot correspond to the strength of the correlation between
variables and community structure. Each circle represents the bacterial community structure for each sample.