# Peer review of "Development of bacterial communities in biological soil crusts along a revegetation chronosequence in the Tengger Desert, northwest China"

_Biogeosciences, 2017_

## Referee Comment (RC1) · Anonymous Referee #1 · 8 May 2017

Development of bacterial communities in biological soil crusts along a revegetation chronosequence in the Tengger Desert, northwest China.

Liu et al.

This manuscript describes the characterization of bacterial communities in shifting sands and in 6 chronological stages of surface stabilization following intervention to reduce dust and soil movement. The authors found that bacterial biodiversity and biomass increased over time after soils were stabilized, reaching relative high points in the 15-years after stabilization samples and staying relatively high through more advanced stages of stabilization and biocrust development. These substantial changes in bacterial community structure are correlated with changes in soil

properties–suggesting a link between biocrust development and rates and quality of soil biogeochemical processes.

Overall, this is a reasonably well-written manuscript that is based on strong methods for assessing bacterial community structure. The manuscript is also nicely divided into focal sections making it easy to follow and understand the results. Questions regarding changes in soil microbial communities over time, particularly as related to succession following disturbances, and what this change in community states mean for ecological functions is a longstanding interest in soil biology. More recently, soil microbial communities have become a focus for testing ecological theory related to community assembly and biodiversity. Thus, the current data set is likely to be of interest to a range of ecologists.

Major Comments:

While the paper has clear strengths as noted above, I have a few major concerns at this point I would like the authors to consider and address. In particular, the current manuscript (while based on high-quality methods) is primarily descriptive and neither focuses on testable hypotheses, nor does it present an advance with regard to our understanding of what controls community assembly and structure across succession in soil microbial communities. Given the great interests in this topic and the many available papers reporting relevant findings in other soil bacterial communities, a better investment and framing of the current results within this context would improve the paper.

On lines 64-67, the authors do present 3 general questions, including questions about the role of BSC bacterial communities in the development of soil properties and roles in ecosystem recovery. However, the current manuscript only addresses the simple question of "what changes occur in bacterial community composition?" In addition, on lines 74-75, the authors hypothesize that bacteria are the key species in C accumulation and soil improvement during BSC succession, but there is no explicit, mechanistic test of

this hypothesis since C accumulation is not parsed among biotic groups and neither is the role of non-BSC soil organisms or simply physical processes in soil improvement.

The authors do a good job of presenting differences in bacterial community structure among successional stages, but the attempt to explain the drivers of these changes is less developed. In particular, Figure 9 shows the results of an RDA which suggest links between soil microbial communities and soil properties. This analysis supports the presence of significant correlations, but it cannot inform readers about what changes first and what is the driver. In other words, we cannot determine to what extent soil properties change and subsequently driver changes in bacterial communities, and to what extent development in bacterial communities drive changes in soil properties. I have 2 main concerns about this section of the manuscript.

First, the soil biogeochemical data used in the RDA appear to have been collected in 2005, 10 years prior to the sampling of the soil bacterial communities. Given likely changes in these soil properties over that 10 year period, the authors need to present this caveat when presenting and discussing their findings. Sampling soil properties in 2005 also make it unclear how soil properties were assessed in the 5-Year category, or how bacterial communities in these samples were compared to soil properties in the RDA behind Figure 9. Where the 5-Year properties gap-filled? A second concern about the RDA approach is the high number of explanatory variables (18) to bacterial samples (21). A key concern in RDA is the potential for 'overdetermination' when a nearly 1:1 ratio of samples to explanatory variables is present. Many of the explanatory variables are sure to be collinear, and over-fitting or over-determining will result. A common method for dealing with this issue is to include an effort to partition the variance explained by each variable in the bacterial community ordinations. It is not uncommon to find that your variables explain >100% of the variation and hence, you are over-fitting the model.

Second, I am concerned about statements made here, and in many other BSC focused papers, about the positive influence of BSCs on soil improvement. At a minimum, con-
sidering that changes from communities in shifting sands to those in later successional stages rely initially on physical processes that stop the movement of soils to allow development of BSCs, it is not safe to say that changes in soil properties in later stages are simply driven by changes in BSC microbes. While many authors have interpreted correlations among soil properties and BSCs as an indicator that BSCs are drivers of soil fertility and development (Zhang et al. 2010, Chamizo et al. 2012, Delgado-Baquerizo 2013, Chen and Duan 2015, Yu et al. 2016), a number of authors have reported the opposite and suggest a direct influence of soil properties on BSC development (Bowker et al. 2006, Rivera-Aquilar et al. 2006, Bowker and Belnap 2008, Root and McCune 2012, Concostrina-Zubiri et al. 2013, Belnap et al. 2014, Bowker et al. 2016). These are important questions and parsing out the interactions of BSCs and soil biogeochemical properties remains an important frontier in BSC research–however, further work to identify controlled experimental approaches are required to answer this question as field correlations leave us wondering about the directionality of controls over time.

---

## Author Comment (AC1) · 1 Jun 2017

I would like to thank Referee #1 for good suggestions and valuation to our manuscript. In this study, we analyzed the soil microbial communities related to succession and want to know a link between biocrust development and soil biogeochemical processes. This is also important to interpret ecological theory related to community assembly and biodiversity, and to explain the mechanism of ecosystem stability maintenance. However, the manuscript has some problems that need to be revised. We think these suggestions are very helpful to improve our manuscript, and the modified sentences or words were marked as red in the revised manuscript. Now we discuss these questions together in the following.

During the experiment and in our writing process, our main goal is to test the hypothesis that bacteria are the main species in C accumulation and soil improvement in early stages of BSC succession. We want to test it by analyzing the bacterial community composition and the potential function of BSCs along a chronosequence in an over 50-year-old revegetation. On lines 64-67, we present three questions in the text. We examined the bacterial community composition, but the other two questions need to be interpreted by existing literature knowledge. Because the bacterial community composition was obtained by high-throughput sequencing, and the experiments to determine concrete functions of each composition could not be realized. So we want to collect the functional information of bacterial community composition at phylum level or genus level to interpret their potential function in the development of soil properties and roles in ecosystem recovery. So the emphasis is in the discussion part. We have added a concluding statement in the discussion part of 'Function of BSC bacteria' (Different compositions of bacterial community play various roles in improving soil properties in different successional stages of BSCs, suggesting their positive potential function in soil biogeochemical cycle and ecosystem process).

On lines 74-75, we hypothesize that bacteria are the key species in C accumulation and soil improvement during BSC succession. In fact, what we really want to say is bacteria have greatest abundance in BSCs compared with other microbial organism such as fungi. So we think they play important roles in C and N accumulation and soil improvement during BSC succession. There was no definite expression in the text and then revised this sentence.

In results of RDA, the referee proposed 2 main concerns. First, the soil biogeochemical data used in the RDA have been collected in 2005. But the data were collected from samples with the same successional stages of our samples in the same experimental site. So we do not think there is a need for repeated analysis of soil physicochemical properties in our samples. The climate of the past 10 years has not changed much and the recovery level of soil and microbe is basically the same in the same development stage. According to the suggestion of "Given likely changes in these soil properties over that 10 year period, the authors need to present this caveat when presenting and discussing their findings", we present this description to the result part of the text as "Taking into account the likely changes in the soil properties from samples with the same successional stages in the same experimental site, we selected soil biogeochemical data collected from 2005 in the RDA (Table S5).". Moreover, the soil biogeochemical data were not actually measured in the 5-Year category, but the fitted curves of all indexes of soil biogeochemical properties were made by the authors. So there is no missing the 5-Year properties. Unfortunately, we have not made this point clear in the text, and no fitting data were shown in Table S5. So we added the fitting data of the sand-fixing sites in Table S5, and the instructions are given in Table S5. Second, in order to dealing with the over-fitting in Fig. 9, we selected 9 presentative variables from 18 of the soil biogeochemical data, and made a new RDA figure. Taking into account the high correlation between variables or the relationship between environmental factors and microbial communities, we selected variables including silt + clay content, water holding capacity, Bulk density, pH, Organic C, Total N, Total P, Total K and electric capacity. The new RDA figure and the result were added into the text.

At the end of the Discussion, we want to state that the microorganisms can in turn improve soil texture. Maybe our language is not accurate enough, we also think that it is not safe to say that changes in soil properties in later stages are simply driven by changes in BSC microbes. So the last sentence have been revised as 'the microorganisms in turn have the positive influence on soil improvement'. Meanwhile, we think the Referee's statement about the interactions of BSCs and soil biogeochemical properties is fits perfectly at the end of our discussion. So we added directly to our manuscript as part of the prospect or vision for the future work in discussion part.

Please also note the supplement to this comment:
http://www.biogeosciences-discuss.net/bg-2017-139/bg-2017-139-AC1- supplement.pdf

**Fig. 1.**

**Supplement:**

[revised manuscript text omitted]

(data from Li et al., 2007a; Table S5). The BSC grouping patterns of bacterial communities at the phylum and genus levels were similar to the OTU level, with all divided into two groups. Group I

contained two members, MS and 5YR, which dominated the physical crusts and cyanobacterial crusts (Figure 1 A and B), and had the lowest diversities with Shannon indexes of 3.3 and 4.61, and

Simpson indexes of 0.139 and 0.0531, respectively (Table S1). The remaining BSCs comprised the largest branch of Group II, which dominated BSCs composed of algae, lichens or mosses (Figure 1

C–F), and had higher diversity with Shannon indexes > 6.0 (Table S1).

From Figure 9, it can be inferred that BSC development was associated with soil physicochemical properties. The development of microbial community structure was positively correlated with the physicochemical index except for soil bulk density. The total variation in OTU

data explained by the first four axes in the RDA (as constrained by the measured environmental variables) was 82.16%, with the first axis explaining 75.27% and the second axis explaining 4.42%.

Of all the environmental factors, silt+clay content and total K were most strongly related to axis 1, with highest correlated variable (silt+clay: -0.91; total K:-0.90). Therefore, silt+clay content and total K were the prime determinants of BSC bacterial community development, shown by the positions of cluster groups along axis 1. Eight soil physicochemical variables were all significant testified by the permutation test analysis ($p < 0.05$): pH; silt and clay content; organic C; total phosphorus (P), nitrogen (N) and potassium (K); electrical conductivity (EC) and water-holding capacity (WHC).

[revised manuscript text omitted]
 have the positive influence on soil improvement (Li et al., 2007b, 2010). Many authors have interpreted correlations among soil properties and BSCs as an indicator that BSCs are drivers of soil fertility and development (Chamizo et al. 2012; Delgado-Baquerizo 2013; Yu et al. 2014; Zhang et al. 2010), a number of authors have reported the opposite and suggest a direct influence of soil properties on BSC development (Bowker et al. 2006, Rivera-Aquilar et al. 2009, Bowker and Belnap 2008, Root and McCune 2012, Concostrina-Zubiri et al. 2013, Belnap et al. 2014, Weber et al. 2016). These are important questions and parsing out the interactions of BSCs and soil biogeochemical properties remains an important frontier in BSC research. However, further work to identify controlled experimental approaches are required to answer this question as field correlations leave us wondering about the directionality of controls over time.

**5 Conclusions**

Assessing of bacterial community structure by Illumina MiSeq sequencing showed that changes of bacterial diversity and richness were consistent with the recovery phase of soil properties in different successional stages of BSCs in the revegetation of Shapotou in the Tengger Desert. The shift of bacterial community composition in BSCs at all levels of classification was related to their corresponding function in the BSC recovery process. These results confirmed our hypothesis that bacteria are important microorganisms in nutrition accumulation and soil improvement in early stages of BSC succession.

*Data availability*. Raw data for Illumina MiSeq sequencing of 18 samples was deposited in the NCBI Sequence Read Archive database (https://www.ncbi.nlm.nih.gov/sra/?term=SRP091312).

*Author contributions*. Lichao Liu and Yubing Liu designed the research. Peng Zhang, Guang Song and Rong Hui collected samples from the field. Yubing Liu and Jin Wang performed DNA extraction and quality detection. Yubing Liu analyzed the high-throughput data and prepared the manuscript with consistent contributions from Lichao Liu. Zengru Wang analyzed the soil biogeochemical data and made the RDA figure.

*Competing interests*. The authors declare that they have no conflict of interest.

*Acknowledgments*. This work was financially supported by the Creative Research Group Program of National Natural Science Foundation of China (grant No. 41621001) and the National Natural Science Foundation of China (grant No. 41371100 and 41401112).

[revised manuscript text omitted]

---

## Referee Comment (RC2) · Anonymous Referee #2 · 11 Jun 2017

This manuscript presents the "Development of bacterial communities in biological soil crusts along a revegetation chronosequence in the Tengger Desert, northwest China". The major findings show that over 60 years, bacterial richness and abundance reached their peak after 15 years of BSC establishment. The results suggested that changes in the bacterial community structure may be an important indicator of the biogeochemical nutrient storage in early successional stages. Overall, I found the information about the bacterial component of BSC presented in the context of succession very interesting and the results in this paper to be sound and well presented. There are very few reports on the bacterial composition of BSC yet their function is equally important to the bigger organisms such as lichens, mosses and cyanobacteria. I commend the authors

for their work and presenting complex data derived from sequencing in a clear manner. There are some suggestions I would recommend to improve the manuscript and to add to depth. General comments Introduction The introduction needs strengthening in terms of explaining the roles of BSC in succession both on the landscape scale and on the micro-scale. Some BSC remain in an early successional state due to the harsh environments that they occupy. This needs to be expanded on. The first paragraph of the introduction was a little repetitive (i.e. saying the same thing twice in a slightly different way) and needs tightening. The last two sentences mention succession in a general sense but then leaps into the role of bacteria in BSC (second paragraph). There have been some very good studies on succession in BSC. If these were brought into the introduction at this point it could help to define the terms of the research more clearly. These studies could highlight the gaps in understanding the role of bacteria in BSC. There are also some relevant points in the first paragraph of the discussion that could be better placed here (see comments in Discussion section). On rereading the paper some points mentioned below are scattered throughout however they need to be brought together to provide a clearer insight into the importance of BSC bacteria and microprocesses in succession. For example: Bowker (2007) examines the role of BSC in primary succession (vs secondary succession) where their role may exist during a time when resources are made available (e.g. light) however they fade into the background once higher vegetation takes over. On the other hand, in some environments of high abiotic stress (e.g. deserts) BSC play a role in succession yet exist as a permanent component. Bowker's review and discussion is supported by work carried out in southern Africa (Büdel et al. 2009) where different successional BSC are described. How do the Tengger Desert BSC differ from the southern African crusts in terms of successional stage? This can be further examined in the discussion of the results in relation to BSC succession and the bacterial component. Büdel et al. (2009) also describes in detail crust types that were representative of successional stages. Do these compare well with your study? This can be further elaborated in the discussion of your results. In Germany, Langhans et al 2009 studies succession at a shorter time scale

with some interesting results. Castillo-Monroy et al (2011) showed few BSC effects on ecosystem function could be ascribed to bacteria. Other literature of interest could include Baran et al (2015) that describes resource partitioning where BSC primary producers (e.g. cyanobacteria) produce an array of metabolites that influence bacterial composition. The hypothesis states that: '. . .bacteria are the key species in C accumulation and soil improvement in the early stages of BSC succession'. In the preceding paragraph N-fixation by heterotrophic bacteria is mentioned yet until the hypothesis C is not discussed. There is no mention of the role of cyanobacteria, cyano-lichens and bacteria and their various roles in C and N fixation or micro-processes (e.g. their role in producing metabolites that are resources for bacteria). These are needed to place this study in the full context of BSC succession and establishment. The key questions may be further developed around: Do bacteria follow the same successional patterns as described in other BSC research? What are the drivers of bacterial composition over time? What are the micro-processes that drive bacterial composition and function? Do bacteria drive changes to soil physicochemical properties or alternatively do the larger BSC organisms drive these changes which in turn has a direct influence on bacterial composition and function? Materials and Methods This section is relatively clear aside from some minor corrections (below). What was the depth of the samples extracted? Line 89 needs further explanation of what 'plantation in floating sand' means. Section 2.2 repeats some descriptions that are also in Figure 1C. This section may be more clearly presented as a table? Results A minor concern with the results is that there is no follow-up description of the key species of the BSC. Section 3.4 touches on the two main groups where the first group is mainly physical and cyanobacterial crusts and the second group is more diverse with algae, lichens and mosses. It would be interesting to have some basic information concerning these organisms in terms of diversity and function so bacterial communities are viewed in that context. This also appears important in terms of succession (also see Büdel, Langhans and Castillo-Monroy papers mentioned and others). You infer that BSC development is associated with bacterial diversity however these results do not identify which crust organisms are drivers of soil

physicochemical properties. The phototrophic organisms of BSC are key C fixers and many also N-fixers. Even though the focus is bacteria the stratification of the processes down the profile is an important part of understanding the role of bacteria and drivers of their diversity. The conclusion that soil physicochemical factors are the driving influence in bacterial diversity and function needs to be qualified especially as the data was used from a much earlier study. With the key changes occurring over the first 15 years it would have been more robust to have run tests. Please be sure that each time algae or algal crusts are mentioned that they are algae and not cyanobacteria as sometimes these terms are interchanged accidently. As with cyanobacteria, lichens and mosses, it would also be good to understand where algae stand in terms of BSC succession in the Tengger Desert. Discussion The first paragraph of the discussion is part introduction material and part discussion that could be better incorporated into discussing with the results. In line 266 algal crusts are referred to – were there algae at the site too? It would be good to be able to refer to some names of the various organisms where applicable. Which lichens and moss appear? Were the lichens N-fixers (cyano-lichens or chloro-lichens). Further down you mention the importance of microbial biomass in terms of succession and their role in conversion rates of organic matter. This information really needs to be in the introduction then discussed in more depth in the results illustrating how your results support this and compares to other studies. I would like to see section 4.1 separated into two paragraphs where the discussion changes between bacteria and cyanobacteria as they differ in their key points (and perhaps other sections looked at too). Here, the knowledge that cyanobacterial crusts are often the permanent state in high stress environments could explain the results (Bowker 2007). The last sentence supports this argument in that these species are well-adapted to drought conditions. Section 4.3 needs depth with a slightly broader discussion where other literature on this topic is referred to in more detail especially in the first part. It is important to clarify how bacteria fit in the BSC picture and why the authors can state that these bacteria are the most likely major contributing microbes in soil physicochemical properties. When describing the recovery phase one would expect the overall diversity of BSC is a crucial factor as well as the influence of the revegetation (secondary succession) that would also provide stability. This is finally mentioned in the last sentence which is too long and loses its impact. Ultimately, it would be good to understand more of the factors that influenced the composition and function after 15 years. Had the vegetation matured? Were abiotic factors (climate) an influence? Had the BSC community composition reached its peak and just ebbed and flowed from this point according to resource availability? These are just some questions that are brought to mind, some of which may have answers and others that need further research in the future. Conclusion The conclusion needs rewriting to summarise the strengths of this manuscript. There are several very interesting results that could be neatly wrapped up including separating out the importance between BSC succession and vegetation succession. On a landscape scale and in high stress environments the role of diversity hot spots of BSC microbes is crucial to establishing stability, regulating moisture and nutrient cycling. Additionally, bacteria are the conduits between the larger BSC organisms and plants facilitating micro-processes. These types of statements will support your final statement that the significant biomass points to bacteria as key contributors to the BSC primary succession process and no doubt in terms of secondary succession as well. I would prefer to see the support of your hypothesis and statements of key findings summarised (as per last paragraph in introduction) in the first paragraph of the discussion. Minor corrections Abstract Line 19 compositional Line 25 bacterial Line 29 Changes. . ..structure Introduction Line 35 BSC or BSCs? Many authors now drop the 's' for Biological Soil Crusts acronym to read BSC even for plural. This sentence needs recasting and perhaps split into two sentences: BSC composition and function. Line 40 missing word – '..landscape xxxx' Line 43 They provide ecological services. . . Line 45 BSC generally exhibit primary successional stages. . .. Line 49 define more clearly 'most abundant' Materials and Methods Line 79/80 natural. . .with a vegetation cover of less than 1%. Line 85 burial. . .dust hazards. . .. Delete 'Also. . .' start sentence with 'The. . ..' Line 87 change to 'fixed sand.' Line 89 delete 's' off literature Line 91 'The BSC could. . ..' Line 94 add (MS) after sand for first mention of

abbreviation Line 99 'In each revegetation site...' add depth of cores Results Line 226 remove space between 31 and % Line 232 delete 'this' Line 245 changed composed to comprised Line 250 change testified to verified Discussion Line 305 '...that display good adaptation to drought conditions together with important roles...' Line 309 More recent information.... Line 344 'properties were...'

---

## Editor Comment (EC1) · A. Antoninka (Editor) · 16 Jun 2017

These are excellent comments and suggestions. The authors have done a good job addressing them.

---

## Editor Comment (EC2) · A. Antoninka (Editor) · 16 Jun 2017

The suggestions are excellent. i look forward to see the authors' responses and revision.

---

## Author Comment (AC2) · 20 Jun 2017

Thank Referee #2 for the good suggestions to improve our manuscript and to add the depth. Now we discuss these questions one by one in the following. Introduction In the introduction, we split BSC composition and function into different paragraghs, and explained the roles of BSC both on the landscape scale and on the micro-scale. Also, we strengthened to explain the roles of BSC in succession in another paragragh. We separated the sentence 'BSC constitute one of the most important landscapes and make up 40% of the living cover of desert ecosystems' from 'It is well known that BSC play critical roles in the structure and function of semi-arid and arid ecosystems'

into two paragraghs. We think these two sentences play the role of up and down convergence in the two paragraghs, so it is not a repeat in such expression this time. According to the suggestions of Referee #2, we placed here the first paragraph of the discussion and adjusted the expression level of BSC roles from landscape scale to the micro-scale and succession. Combined with the full text of the expression, we deleted the hypothesis that bacteria play important roles in carbon (C) accumulation and soil improvement in early stages of BSC succession. Meanwhile, we added Section 4.4 for discussion the role of BSC to succession. Materials and Methods In our study, we selected BSC layer from the revegetation of different successional stages. Thus the depth of BSC was varied in these fixed-sand areas of different years (the depth was shown in Table S5). The whole BSC layer were selected with different depth in different sampling sites. In Line 89, 'plantation in floating sand' means 'plantation of the xerophilous shrubs in mobile sand'. But this meaning was expressed in the sentence 'The revegetation protection system for Bao–Lan railway in this area was established initially in 1956, and was expanded in 1964, 1973, 1981 and later through the plantation of the xerophilous shrubs'. So we decided to delete 'plantation in floating sand' in this sentence. Yes, in Section 2.2, there are some repeated descriptions, we have deleted and revised them. This section may be more clearly presented as a table. We think such statements can clearly demonstrate our sampling steps, so we didn't prepare a table in this section. Results In the results, the description of the key species of BSC was in paragragh 3, section 3.3. In Section 3.4 and 2.1, the BSC types were in unified terms as physical crusts, algal-dominated, lichen-dominated and moss-dominated crusts in the revised manuscript. Algae or algal crusts are mentioned and not cyanobacteria in the text. In the results of RDA, Both Referee #1 and Referee #2 put forward valuable opinions. The conclusion was revised as soil physicochemical factors are closely related (not determined) to bacterial diversity and function. The revised detail and answer to reviewer were in the revised manuscript and answer to Referee #1. Discussion We have move the first paragraph of the discussion with part introduction material to the Introduction. The first succession stage of BSC was revised as algal crusts in the whole text, thus we continued use the expression 'algal crusts' in line 266. Due to the mixed components of each BSC samples, we can not refer to some names of the various organisms (e.g. which lichens and moss appear? Were cyano-lichens or chloro-lichens?). The importance of microbial biomass in terms of succession and their role in conversion rates of organic matter have been mentioned in Introduction, and discussed the results compared to other studies. We think Section 4.1 can not be separated into two paragraphs. Because this part concerns information of the whole bacterial community compositions, and comparisons with other studies of BSC and elaboration of their functions. Section 4.2 have been separated into two paragraphs. In Section 4.3, a slightly broader discussion on the correlations between soil properties and BSC was added according to suggestions of Referee #1. The predominant bacteria have the largest relative abundance, so they are the most likely major contributing microbes in soil physicochemical properties. The overall diversity of BSC has been added in the factors because it is a crucial factor as well as the influence of the revegetation (secondary succession) that would also provide stability. Finally, the last sentence has been revised as 'It would be good to understand more of the factors that together influenced the composition and function of BSC bacteria in long-term revegetation, including BSC, plant, soil biochemical properties and climate conditions, and the microorganisms in turn have the positive influence on soil improvement'. We added Section 4.4 for discussion the role of BSC to succession. Conclusion The conclusion have been rewritten and strengthened the contribution to the BSC succession process. Also, the support of our hypothesis and statements of key findings have been summarised in the first paragraph of the discussion. Minor corrections All minor corrections that suggested by Referee #2 have been revised in the revised manuscript.

Please also note the supplement to this comment:
http://www.biogeosciences-discuss.net/bg-2017-139/bg-2017-139-AC2-supplement.pdf

[Figure]

**Supplement:**

**Development of bacterial communities in biological soil crusts along a revegetation chronosequence in the Tengger Desert, northwest China**

***Author names and affiliations:***

Lichao Liu[1], Yubing Liu[1, 2]*, Peng Zhang[1], Guang Song[1], Rong Hui[1], Zengru Wang[1], Jin Wang[1, 2]

[1]Shapotou Desert Research & Experiment Station, Northwest Institute of Eco-Environment and Resources, Chinese Academy of Sciences, Lanzhou, 730000, China

[2]Key Laboratory of Stress Physiology and Ecology in Cold and Arid Regions of Gansu Province, Northwest Institute of Eco–Environment and Resources, Chinese Academy of Sciences, Lanzhou 730000, China

*****Corresponding author:*** Yubing Liu

Address: Donggang West Road 320, Lanzhou 730000, P. R. China.

Tel: +86 0931 4967202.

E-mail address: liuyb@lzb.ac.cn

**Abstract.** Knowledge of structure and function of microbial communities in different successional stages of biological soil crusts (BSC) is still scarce for desert areas. In this study, Illumina MiSeq sequencing was used to assess the compositional changes of bacterial communities in different ages of BSC in the revegetation of Shapotou in the Tengger Desert. The most dominant phyla of bacterial communities shifted with the changed types of BSC in the successional stages, from Firmicutes in mobile sand and physical crusts to Actinobacteria and Proteobacteria in BSC, and the most dominant genera shifted from *Bacillus*, *Enterococcus* and *Lactococcus* to RB41_norank and JG34-KF-361_norank. Alpha diversity and quantitative real-time PCR analysis indicated that bacterial richness and abundance reached their highest levels after 15 years of BSC development. Redundancy analysis showed that silt+clay content and total K were the prime determinants of the bacterial communities of BSC. The results suggested that bacterial communities of BSC recovered quickly with the improved soil physicochemical properties in the early stages of

BSC succession. Changes in the bacterial community structure may be an important indicator in the biogeochemical cycling and nutrient storage in early successional stages of BSC in desert ecosystems.

**Key words** biological soil crusts (BSC), successional stages, bacterial community, revegetation, desert ecosystem

**1 Introduction**

[revised manuscript text omitted]

Bowker (2007) examines the role of BSC in primary succession (vs secondary succession) where their role may exist during a time when resources are made available (e.g. light). However, they fade into the background once higher vegetation takes over. On the other hand, in some environments of high abiotic stress (e.g. deserts), BSC play a role in succession yet exist as a permanent component. Bowker's review and discussion is supported by work carried out in southern Africa (Büdel et al., 2009) where different successional BSC are described. Büdel et al. (2009) also describes in detail crust types that were representative of successional stages. Castillo-Monroy et al. (2011) showed few BSC effects on ecosystem function could be ascribed to bacteria.

A recent study on crusts in the Tengger Desert, China, showed that bacterial diversity and richness were highest after 15 years, and at least 15 years might be needed for recovery of bacterial abundance of BSC (Liu et al., 2017). To better understand these questions, we must analyze in detail the bacterial community composition of BSC at all levels of classification and their corresponding function in the recovery process of BSC. In the present study, bacterial community composition and potential function were analyzed in BSC along a chronosequence of over 50-year-old revegetation. We want to know: what are the drivers of bacterial composition over time? What are the micro-processes that drive bacterial composition and function? Do bacteria drive changes to soil physicochemical properties or alternatively do the larger BSC organisms drive these changes which in turn has a direct influence on bacterial composition and function?

[revised manuscript text omitted]
. Taking into account the likely changes in the soil properties from samples with the same successional stages in the same experimental site, we selected soil biogeochemical data collected from 2005 in the RDA (data from Li et al., 2007a; Table S5). The BSC grouping patterns of bacterial communities at the phylum and genus levels were similar to the OTU level, with all divided into two groups. Group I contained two members, MS and 5YR, which dominated the physical crusts and algal crusts (Figure 1 A and B), and had the lowest diversities with Shannon indexes of 3.3 and 4.61, and Simpson indexes of 0.139 and 0.0531, respectively (Table S1). The remaining BSC comprised the largest branch of Group II, which dominated BSC composed of algae, lichens or mosses (Figure 1 C–F), and had higher diversity with Shannon indexes > 6.0 (Table S1).

From Figure 9, it can be inferred that BSC development was associated with soil physicochemical properties. The development of microbial community structure was positively correlated with the physicochemical index except for soil bulk density. The total variation in OTU data explained by the first four axes in the RDA (as constrained by the measured environmental variables) was 82.16%, with the first axis explaining 75.27% and the second axis explaining 4.42%. Of all the environmental factors, silt+clay content and total K were most strongly related to axis 1, with highest correlated variable (silt+clay: -0.91; total K:-0.90). Therefore, silt+clay content and total K were closely related to bacterial community development of BSC, shown by the positions of cluster groups along axis 1. Eight soil physicochemical variables were all significant verified by the permutation test analysis ($P < 0.05$): pH; silt and clay content; organic C; total phosphorus (P), nitrogen (N) and potassium (K); electrical conductivity (EC) and water-holding capacity (WHC).

**3.5 Quantification of bacterial abundance**

The averaged bacterial abundance in MS was $1.12 \times 10^6$ copies (16S rRNA gene) per gram of soil (Table 2). Similar to the shift of bacterial richness, gene copies increased quickly in the initial 15 years of BSC development, and reached the approximate highest level of $2.70 \times 10^8$ copies in 15YR. There were no significant differences among 28YR, 34YR and 51YR.

**4 Discussion**

On a landscape scale and in high stress environments, the role of diversity hot spots of BSC microbes is crucial to establishing stability, regulating moisture and nutrient cycling (Bowker, 2007). Additionally, bacteria are the conduits between the larger BSC organisms and plants facilitating micro-processes (Castillo-Monroy et al., 2011), and thus bacteria as key contributors to the BSC primary succession process and no doubt in terms of secondary succession as well.

**4.1 Impact of BSC age on bacterial community composition**

[revised manuscript text omitted]

*Blastococcus* and *Arthrobacter* in the present study. Different compositions of bacterial community play various roles in improving soil properties in different successional stages of BSC, suggesting their positive potential function in soil biogeochemical cycle and ecosystem process. Further discovery and characterization of the functions of these dryland-adapted bacteria is a challenging area for future study.

**4.3 Relationship between bacterial community shift and soil physicochemical properties**

PCA and RDA showed that bacterial community compositions of MS and 5YR significantly differed from those of BSC of more than 15 years in age, and were positively correlated with soil physicochemical properties. Combined with the results of alpha-diversity analysis and qPCR, this means that the species richness and abundance reached their highest levels at 15 years of BSC development and then maintained similar levels thereafter. Similar trends were found in recovery of soil properties and processes after sand-binding at five different-aged revegetated sites – proportions of silt and clay, organic C increased with years since revegetation (Li et al., 2007a, b). The annual recovery rates of soil properties were greater in the initial revegetated sites (0–14 years) than that in the old revegetated sites (43–50 years) (Li et al., 2007a). These results suggest that bacterial communities of BSC recovered quickly in the fastest recovery phase of soil properties (the initial 15 years), and the bacterial biomass increased with the improvement of soil texture and nutrients, especially silt, clay and total K content in the Tengger Desert. A significant positive correlation was found between silt and clay and the number of BSC types in southern Africa (Büdel et al., 2009), suggesting that fine grain-size promotes BSC succession and their biomass content. This may be attributed to the diversity of BSC, vegetation composition, soil temperature and soil moisture. Because they are key factors regulating soil microbial composition and activity (Butenschoen et al., 2011; De Deyn et al., 2009; Sardans et al., 2008), soil nutrient uptake and release (Peterjohn et al., 1994; Rustad et al., 2001), especially in the BSC of top soil. It would be good to understand more of the factors that together influenced the composition and function of BSC bacteria in long-term revegetation, including BSC, plant, soil biochemical properties and climate conditions, and the microorganisms in turn have the positive influence on soil improvement (Li et al., 2007b, 2010).

Many reports have interpreted correlations among soil properties and BSC as an indicator that BSC are drivers of soil fertility and development (Chamizo et al., 2012; Delgado-Baquerizo

2013; Yu et al., 2014; Zhang et al., 2010), some have reported the opposite and suggest a direct influence of soil properties on BSC development (Bowker et al., 2006, Rivera-Aquilar et al.,

2009, Bowker and Belnap 2008, Root and McCune 2012, Concostrina-Zubiri et al., 2013,

Belnap et al., 2014, Weber et al., 2016). These are important questions and parsing out the interactions of BSC and soil biogeochemical properties remains an important frontier in BSC

research. However, further work to identify controlled experimental approaches are required to answer this question as field correlations leave us wondering about the directionality of controls over time.

**4.4 The role of BSC to succession**

In temperate desert regions, BSC are not well investigated regarding community structure and diversity. Furthermore, studies on succession are rare (Langhans et al., 2009). Most evidence indicates that BSC facilitate succession to later seres, suggesting that assisted recovery of BSC could speed up succession (Bowker, 2007). Because BSC are ecosystem engineers in high abiotic stress systems, loss of BSC may be synonymous with crossing degradation thresholds. Whether or not

BSC are deemed facilitative or inhibitory for later successional vegetation may depend on how exhaustively the interaction between plants and BSC. On fixed-sand areas, BSC may in some cases reduced infiltration (inhibitory effect) (Mitchell et al., 1998), but they also increased soil stability and served as an N source for surviving and recolonizing trees (facilitative effects) (Uchida et al.,

2000; Tateno et al., 2003). BSC bacterial communities recovered in the successional stages may help establishing stability, regulating nutrient and biogeochemical cycling. Castillo-Monroy et al.

(2011) found that BSC richness matrix has the greatest direct effect on the ecosystem function matrix. Despite this result, very few of the BSC effects on ecosystem function could be ascribed to changes within the bacterial community. It provides valuable insights on semi-arid ecosystems where plant cover is spatially discontinuous and ecosystem function in plant interspaces is regulated largely by BSCs.

**5 Conclusions**

Illumina MiSeq sequencing showed that changes of BSC bacterial diversity and richness in BSC

succession were consistent with the recovery phase of soil properties in vegetation succession of Shapotou in the Tengger Desert. The shift of bacterial community composition in BSC at all levels of classification was related to their corresponding function in the BSC recovery process. BSC bacteria are crucial to establish stability and nutrient cycling in desert ecosystem, and are the conduits between the larger BSC organisms and plants facilitating micro-processes. These results confirmed our hypothesis that bacteria as key contributors to the BSC succession process.

*Data availability*. Raw data for Illumina MiSeq sequencing of 18 samples was deposited in the NCBI Sequence Read Archive database (https://www.ncbi.nlm.nih.gov/sra/?term=SRP091312).

*Author contributions*. Lichao Liu and Yubing Liu designed the research. Peng Zhang, Guang Song and Rong Hui collected samples from the field. Yubing Liu and Jin Wang performed DNA extraction and quality detection. Yubing Liu analyzed the high-throughput data and prepared the manuscript with consistent contributions from Lichao Liu. Zengru Wang analyzed the soil biogeochemical data and made the RDA figure.

*Competing interests*. The authors declare that they have no conflict of interest.

*Acknowledgments*. This work was financially supported by the Creative Research Group Program of National Natural Science Foundation of China (grant No. 41621001) and the National Natural Science Foundation of China (grant No. 41371100 and 41401112).

[revised manuscript text omitted]

34YR and 51YR represent mobile sand, 5-, 15-, 28-, 34- and 51-year-old BSC, respectively.

[Figure]

**Figure 5**. Abundant phyla (> 10% of total OTUs) and low-abundance phyla (1% < of total OTUs < 10%) of bacteria distributed in different ages of BSC. Data are defined at a 3% OTU genetic distance. Data are presented as mean ±

standard deviation; n = 3 per BSC sample. Paired t-test (BSC samples) was used to assess the significance between adjacent ages of BSC. *P ≤ 0.05, **P ≤ 0.001. MS, 5YR, 15YR, 28YR, 34YR and 51YR represent mobile sand, 5,

15, 28, 34 and 51-year-old BSC, respectively.

[Figure]

**Figure 6**. Bacterial community composition in six different ages of BSC at the genus level. Data are defined at a 3%

OTU genetic distance. MS, 5YR, 15YR, 28YR, 34YR and 51YR represent mobile sand, 5, 15, 28, 34 and 51-year- old BSC, respectively.

[Figure]

**Figure 7**. Phylogenetic relationship of the 30 most abundant genera in bacterial composition of BSC.

[Figure]

**Figure 8**. Abundant species (> 10% of total OTUs) and low-abundance species (1% < of total OTUs < 10%) of bacteria distributed in different ages of BSC. Data are defined at a 3% OTU genetic distance. Data are presented as mean ± standard deviation; n = 3 per BSC samples; Paired t-tests (BSC samples) were used to assess the significance between the adjacent ages of BSC. *$P \leq 0.05$, **$P \leq 0.001$. MS, 5YR, 15YR, 28YR, 34YR and 51YR represent mobile sand, 5, 15, 28, 34 and 51-year-old BSC, respectively.

[Figure]

**Figure 9**. Redundancy analysis (RDA) of bacterial community structures in relation to soil physiochemical properties. Arrows indicate the direction and magnitude of soil physiochemical index associated with bacterial community structures. The length of arrows in the RDA plot correspond to the strength of the correlation between variables and community structure. Each circle represents the bacterial community structure for each sample.

---

## Author Comment (AC3) · 28 Jun 2017

The revised supplement tables.

Please also note the supplement to this comment:
https://www.biogeosciences-discuss.net/bg-2017-139/bg-2017-139-AC3-supplement.zip